# Mountain sickness in altitude inhabitants of Latin America: A systematic review and meta-analysis

J. Pierre Zila-Velasque[1], Pamela Grados-Espinoza[1], P. Alejandra Goicochea-Romero[1,2], Gustavo Tapia-Sequeiros[1,3], J. Enrique Pascual-Aguilar[1], Arturo J. Ruiz-Yaringaño[1,4,5], Shamir Barros-Sevillano[6], Jhon Ayca-Mendoza[1], Wendy Nieto-Gutierrez[7] *

1 Red Latinoamericana de Medicina en la Altitud e Investigación (REDLAMAI), Pasco, Peru, 2 Facultad de Ciencias de la Salud, Carrera de Medicina Humana, CHANGE Research Working Group, Universidad Científica del Sur, Lima, Peru, 3 Facultad de Ciencias de la Salud, Universidad Privada de Tacna, Tacna, Peru, 4 Sociedad Científica de San Fernando, Lima, Peru, 5 Facultad de Medicina Humana, Universidad Nacional Mayor de San Marcos, Lima, Peru, 6 Facultad de Ciencias de la Salud, Escuela de Medicina, Universidad César Vallejo, Trujillo, Perú, 7 Unidad de Investigación para la Generación de Síntesis de Evidencia en Salud, Vicerrectorado de Investigación, Universidad San Ignacio de Loyola, Lima, Peru

* wendy_nieto22@hotmail.com

**Data Availability Statement:** All data used for the study has been included in the manuscript and supplementary material. Supplementary material

## Abstract

### Objective

Chronic and acute mountain sickness is known worldwide, but most of the available information comes from the eastern continent (Himalayas) without taking into account the west which has the most recent group located at altitude, the Andes. The aim of this study was to synthesize the evidence on the prevalence of acute and chronic mountain sickness in Latin American countries (LATAM).

### Methods

A systematic search of the variables of interest was performed until July 8, 2023 in the Web of Science, Scopus, PubMed and Embase databases. We included studies that assessed the prevalence of mountain sickness in high-altitude inhabitants (>1500 m.a.s.l) who lived in a place more than 12 months. These were analyzed by means of a meta-analysis of proportions. To assess sources of heterogeneity, subgroup analyses and sensitivity analyses were performed by including only studies with low risk of bias and excluding extreme values (0 or 10,000 ratio). PROSPERO (CRD42021286504).

### Results

Thirty-nine cross-sectional studies (10,549 participants) met the inclusion criteria. We identified 5 334 and 2 945 events out of 10,000 with acute and chronic mountain sickness in LATAM countries. The most common physiological alteration was polycythemia (2,558 events), while cerebral edema was the less common (46 events). Clinical conditions were more prevalent at high altitudes for both types of MS.

associated with this article can be found in the online version at doi: 10.6084/m9.figshare. 24501796.

**Funding:** The author(s) received no specific funding for this work.

**Competing interests:** The authors have declared that no competing interests exist.

## Conclusion

Acute mountain sickness (AMS) occurs approximately in 5 out of 10 people at high altitude, while chronic mountain sickness (CMS) occurs in 3 out of 10. The most frequent physiological alteration was polycythemia and the least frequent was cerebral edema.

## Introduction

Latin America (LATAM) is home to the world's most extended mountain range, which traverses Argentina, Bolivia, Chile, Colombia, Ecuador, Peru and part of Venezuela. This mountain range boasts an average altitude ranging from 1,500 to over 4,000 meters above sea level (m) [1]. Consequently, LATAM is inhabited by approximately 40 million people who reside at high altitudes, with over 5 million of them living at altitudes exceeding 4,000 m. [1, 2], making them susceptible to mountain sickness (MS).

MS is a syndrome affecting both the cerebral and pulmonary systems and occurs due to hypoxia following an initial ascent to higher altitudes [3]. The diagnosis is based on clinical evaluation, requiring the identification of characteristic hypoxic symptoms. However, these symptoms can vary in severity [4], potentially leading to conditions like pulmonary and cerebral edema, polycythemia, hemorrhage, ataxia, and even coma in some cases [5–7], and depend on the duration of the exposure [5].

Previous studies have reported the frequency of MS in different regions of the world [8–10]. However, these reports are mainly from eastern areas (such as Nepal Himalaya and Ethiopian highlands) with comparatively scarce information with other mountainous places in the world, such as the Andes [11]. In fact, a previous systematic review [12] determined a global prevalence of MS at approximately 12%; however, they omitted to include studies from LATAM despite being from regions with cities over 4000 m. with residents susceptible to MS [13–15].

Likewise, none of the reports describe the frequencies of MS according to the types of symptoms (mild and severe), exposure (acute or chronic), and complications (exaggerated pulmonary hypertension, pulmonary and cerebral edema). The absence of this information can result in the mishandling of data and a lack of appropriate solutions for managing symptoms and allocating health resources efficiently, in addition to the fact that little research from the Andean countries is evident [16]. Therefore, the present study aims to synthesize the evidence on the prevalence of acute and chronic MS in Latin American countries.

## Methods

### Protocol registration and research question

A systematic review was performed followed the guidelines of the Preferred Reporting Items for Systematic Reviews and Meta-Analyses 2020 (PRISMA) (S1 Checklist) [17] and the Cochrane Handbook [18]. The protocol of the study was registered in the International prospective register of systematic reviews—PROSPERO (CRD42021286504) [19].

The research question was "*What is the prevalence of Mountain Sickness in Altitude Inhabitants of Latin America?*", with the following PECO (where the means of P: population E: exposure, O: outcome) structure: 1) population: altitude inhabitants (defined as a person who live in a place more than 12 months); 2) exposure: altitude (following the ranges of intermediate altitude 1500–2500 m, from where physiological changes are detectable), high altitude (2500–3500 m), very high altitude (3500–5800 m), extreme altitude (>5800 m.), and "death zone"

(>8000 m.a.s.l.) [20]; and 3) outcome: prevalence of AMS and CMS, according to each disease and study.

## Systematic search

A systematic search was performed up to July 8, 2023 in the databases of Web of Science (WoS), Scopus, Medline (thought PubMed) and Embase. We adapted the search strategy for each of the databases using keywords related to "mountain sickness" and "Latin America". Also, we include terms for each Latin America country that report a high altitude (Peru, Chile, Argentina, Bolivia, Venezuela, Ecuador, and Colombia), taking into account all the cities located at high altitudes. The search strategy of each database is available in S2 Table. Also, we evaluated the references of included studies to identify potentially eligible studies that weren't found in the systematic search.

## Study selection and data extraction

Duplicates were removed by two independent reviewers using Endnote v.20. The study selection was performed by four blinded and independent reviewers (JEPA, AJRY, PAGR, GTS) divided into two groups of two. The selection was performed in two stages, the first, a selection to evaluate the title and abstract, and the second, to evaluated the full-text. If there were discrepancies, these were resolved by consensus. The selection process was described in Fig 1.

We included observational studies (cross-sectional or cohort studies) that report the prevalence of MS in people living at altitude in some Latin America country. We excluded editorials, commentaries, opinions, congress abstract, and reviews. No publication date or language restrictions were applied. The studies that were excluded and the reason for exclusion are described in S1 Table.

The data extraction was performed by four independent authors (JEPA, AJRY, PAGR, GTS) divided into two groups of two. We extracted variables like the first author, year of publication, study design, country, sample size, age, sex, type of resident (native: defined as a person who has been born and raised at altitude, or resident: defined as a person who leaves his or her

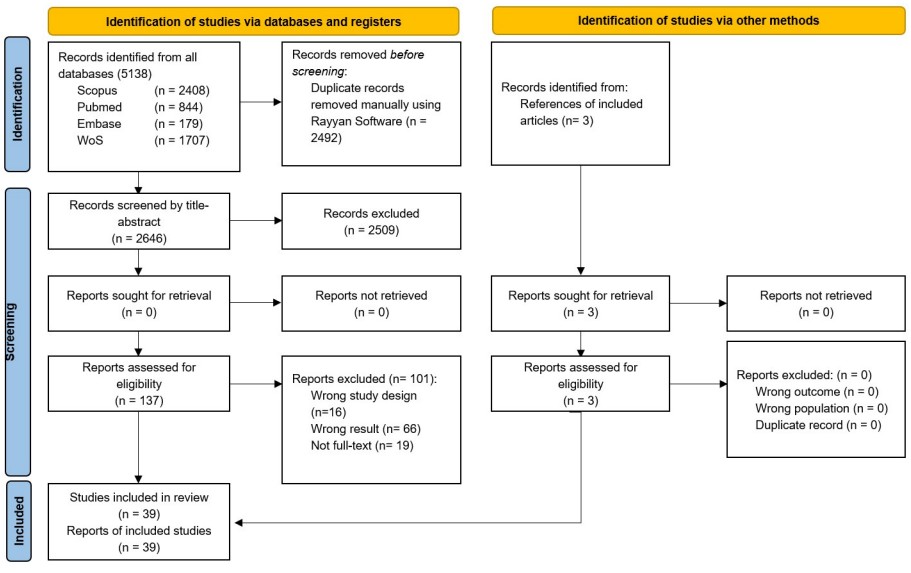

**Fig 1. PRISMA flow chart for the study selection process.**

city of birth to settle elsewhere and who at altitude has a period of stay of more than 12 months), altitude level, time of residence, place of residence (urban: defined like a place that has at least 100 houses together forming blocks or blocks and includes population centers that are district capitals and rural: defined as a place that does not meet the criteria of an urban area [21]), time of exposure, sample of people with acute and chronic mountain sickness, number of events, and type of symptoms. The discrepancies were resolved by a consensus.

## Risk of bias

Four authors (PGE, PAGR, AJRY, JAM) independently assessed the methodological quality of prevalence studies using the Joanna Briggs Institute Critical Appraisal Tool [22]. This scale has 9 items (1, Was the sample frame appropriate to address the target population? 2. Were study participants sampled appropriately?, 3. Was the sample size adequate?, 4. Have the study subjects and environment been described in detail?, 5. Has the data analysis been performed with sufficient coverage of the identified sample?, 6. Were valid methods used to identify the condition?, 7. Was the condition measured in a standard and reliable way for all participants?, 8. Was there adequate statistical analysis?, 9. Was the response rate adequate and, if not, was the low response rate adequately managed?), with possible answers of "Yes", "No" and "Unclear". The quality score was considered as one point for "Yes" and zero points for "No" and "Unclear". We estimated the overall risk of bias by considering the lowest risk of bias reported among the items (S1 Fig).

## Synthesis of the evidence

We used summary statistics to describe the studies, subjects, and outcomes. For the meta-analysis, we pooled the proportions (events over the total population) using a generalized linear mixed random model. To stabilize the variances, we applied the Freeman-Tukey Double Arcsine transformation, and we estimated the confidence intervals for individual study results using the Clopper-Pearson method. We assessed the presence of heterogeneity between studies through visual inspection of forest plots for all outcomes and supplemented this with an evaluation of the $I^2$ parameter. The proportions were expressed per 10,000 population. The bias of publication was evaluated using the graphic of funnel plot.

As a priori subgroup analysis, we explored the proportion of MS across different altitudes, countries, types of symptoms (mild or severe), and complications (pulmonary edema, cerebral edema, exaggerated pulmonary hypertension, and exaggerated polycythemia). These measured outcomes were extracted as reported in the included studies, which were likely defined according to established international criteria. Sensitivity analysis was performed by including only studies with a low risk of bias and excluding extreme values (0 or 10,000 proportion). All analyses were conducted using R Studio.

## Results

### Characteristic of the studies

After eliminating duplicates, we identified 2,492 articles, of which 139 were evaluated in full text, and only 39 studies were included. The studies included evaluated a sample population of 10 549 with an altitude ranging from 2 640 in Colombia [23], to 6 487, in Argentina [24]. We did not identified studies from Brazil, Costa Rica, Cuba, Guatemala, Haiti, Honduras, Dominican Republic and Panama. Of the 39 studies, acute mountain sickness (AMS) was evaluated in 15 studies [24–38] (n = 2 945), chronic mountain sickness (CMS) in 22 [13, 23, 39–58] (n = 7 448), and both in 2 [59, 60] (n = 156). Regarding the studies that evaluated AMS, most of the

studies were from Chile (n = 8 studies) and Peru/Argentina (n = 4 studies), which are the countries that contributed the most research on the subject. There was an altitude ranging from 2 700 m. [31] to 6 487 m [24]. On the other hand, most of the studies that analyzed CMS were from Peru (n = 18 studies), with an altitude ranging from 2 240 m. [51] to 5 300 m. [13] (Table 1).

## Prevalence of mountain sickness

We identified 39 studies which it has the necessary information for analyzed the overall prevalence. We identified that in 10 000 people, 5 334 reported AMS (95% CI: 3 780–6 857, I2 = 97.0%, n = 2 989) and 2 945 reported CMS (95% CI: 1 795–4 237, I2 = 98.0%, n = 7 518) (Fig 2).

The number of events of AMS by 10 000 people according to the levels of altitude the events were 5 728 (95% CI: 2 497–8 659, I2 = 94.0%, n = 1 067–4 studies) at high altitude, 5 323 (95% CI: 3 339–7 258, I2 = 96.0%, n = 1 027–12 studies) at very high altitude, and 4 179 (95% CI: 3 853–4 509, n = 895–1 study) at extreme altitude. In relation to CMS, we identified that the totally of people had symptoms at intermediate altitude (n = 57) and high altitude (n = 6). However, 2 520 (95% CI: 1 609–3 551, I2 = 97.0%, n = 7 455–21 studies) events of CMS were reported in high altitude in a population of 10 000 persons (Fig 3).

We included 25 and 21 studies to mild (headache) and severe (pulmonary and cerebral edema, pulmonary hypertension and polycythemia) symptoms, which were reported 3 932 (95% CI: 2 678–5 259, I2 = 98.0%, n = 9 569) and 1 776 (95% CI: 675–3 220, I2 = 99.0%, n = 9 658) that in 10 000 persons, respectively (Fig 4). According to the order of frequency, we found that polycythemia was most frequent with 2 558 events (95% CI: 1 151–4 278, I2 = 99.0%, n = 6 831–14 studies), followed by pulmonary hypertension with 2 035 (95% CI: 0. 00–5 825, I2 = 96.0%, n = 149–4 studies), pulmonary edema with 1 124 (95% CI: 0.00–5 320, I2 = 91.0%, n = 1 181–5 studies) and finally cerebral edema was present in 46 events (95% CI: 1.14–129, I2 = 7.0%, n = 982–2 studies) out of 10 000 persons in all the aforementioned pathologies (Fig 5).

In relation to the differences by country, we found that a highest prevalence of AMS was in Peru (6 171 events per 10 000 persons, CI 95% 3 628–8 425, I2 = 94%, n = 4 studies) and Chile (5 896 events per 10 000 persons, CI 95% 3 619–7 996, I2 = 95%, n = 9 studies), and for CMS was in Mexico (10 000 events per 10 000 persons, CI 95% 9 701–10 000, n = 1 study) and Colombia. (3 333 events per 10 000 persons, CI 95% 130–7 636, I2 = 95%, n = 1 study) (Fig 6).

## Risk bias assessment, publication bias, and sensitivity analysis

We assessed a high risk of bias in the overall included studies and for each outcome individually. Only three studies [28, 38, 45] are considered to have a low risk of bias in all domains, and only one [43] is classified as having an unclear risk. The domain with the highest risk was "Was the sample size adequate?" followed by "Was the response rate adequate and, if not, was the low response rate adequately managed?". On the other hand, the domain with the lowest risk was "Have the study subjects and environment been described in detail?" (S1 Fig).

The studies with low risk of bias shows a lower prevalence of AMS (4 134 events in 10 000; 95% CI 3 836–4 437 vs 5 334 events in 10 000; 95% CI 3 780–6 857) and CMS (1 621 events in 10 000; 95% CI: 1 310–1 971 vs 2 945 events in 10 000; 95% CI: 1 795–4 237) compared with main analysis; however, their confidence intervals overlap. (S2 Fig). In the same way, the sensitivity analysis excluding the outliers shows a lower prevalence of AMS (4,526 events in 10,000; 95% CI 3,294–5,787 vs. 5,334 events in 10,000; 95% CI 3,780–6,857) and CMS (2,533 events in 10,000; 95% CI: 1,637–3,542 vs. 2,945 events in 10,000; 95% CI: 1,795–4,237) compared with the main analysis, but with overlapping confidence intervals. (S3 Fig).

**Table 1. Characteristics of studies assessing the prevalence of acute and chronic mountain sickness in Latin America.**

| Author-Year | Study design | Country | Altitude level (m) | Type to exposure | Sample size (n) | Age (yr) | Female (%) | Native (%) | Urban (%) |
|---|---|---|---|---|---|---|---|---|---|
| Appenzeller O, et al. 2004 [41] | Prospective cohort | Peru | 4338 | Chronic | 31 | 42.2 ± 1.5* | - | 100 | - |
| Appenzeller O, et al. 2006 [40] | Prospective cohort | Peru | 4338 | Chronic | 9 | 36.9 ± 2.8* | 0 | 100 | - |
| Bilo G, et al. 2020 [42] | Cross-sectional | Peru | 4340 | Chronic | 289 | 38.3 ± 13.2* | 49.5 | 100 | - |
| Brito J, et al. 2007 [59] | Cross-sectional | Chile | 3550 | Both | | 48.7 ± 2* | 0 | 50 | - |
| Brito J, et al. 2018 [29] | Cross-sectional | Chile | 4400–4800 | Acute | | 41.8 ± 0.7* | 0 | 100 | - |
| Cabello G, et al. 2017 [30] | Prospective cohort | Chile | 3500 | Acute | | NR | 12 | - | - |
| Caravedo MA, et al. 2022 [38] | Prospective cohort | Peru | 3350 | Acute | | 21 (20–25) a | 57.0 | 0 | 100 |
| De Ferrari A, et al. 2014 [43] | Cross-sectional | Peru | 3825 | Chronic | 1065 | 55.3 ± 12.6* | 51 | 0 | 49 |
| Garófilo A, et al. 2010 [31] | Prospective cohort | Argentina | 2700–4300 | Acute | | 34 (18–50) ** | 1.59 | - | - |
| Gazal S, et al. 2019 [44] | Prospective cohort | Peru | 4380 | Chronic | 312 | 46.8 ± 13.4* | 0 | 100 | - |
| Gonzales GF, et al.1998 [32] | Prospective cohort | Peru | 3400 | Acute | | 21–30 b | 0 | - | - |
| Gonzales GF, et al. 2009 [46] | Prospective cohort | Peru | 4340 | Chronic | 41 | 47.3 ± 1.2* | 0 | 100 | - |
| Gonzales, GF, et al. 2011 [47] | Cross-sectional | Peru | 4340 | Chronic | 41 | 44.7 ± 9.3* | 0 | 100 | - |
| Gonzales GF, et al. 2011 [58] | Cross-sectional | Peru | 4340 | Chronic | 103 | 45.4 ± 0.8* | 0 | 100 | - |
| Gonzales GF, et al. 2013 [45] | Cross-sectional | Peru | 4100 | Chronic | 506 | 35–75 b | 69 | - | - |
| Hancco I, et al. 2020 [13] | Cross-sectional | Peru | 5100–5300 | Chronic | 1594 | 32 (23–39) a | 14.7 | 0 | - |
| Irarrazaval S, et al. 2017 [33] | Prospective cohort | Chile | 3920 | Acute | | 30.1 (24.9–32.7) a | 0 | - | - |
| Jefferson JA, et al. 2004 [60] | Cross-sectional | Peru | 4300 | Both | | 36.6 (23–58) ** | - | 0 | 100 |
| Lang M, et al. 2021 [34] | Prospective cohort | Chile | 3300 | Acute | | 12.5 ± 1.1* | 57.1 | - | 100 |
| Leon-Velarde F, et al. 1993 [52] | Cross-sectional | Peru | 4300 | Chronic | 2875 | 20–69 b | - | 0 | - |
| Leon-Velarde F, et al. 1994 [53] | Cross-sectional | Peru | 4300 | Chronic | 97 | NR | 0 | 87 | - |
| Leon-Velarde F, et al. 1997 [55] | Cross-sectional | Peru | 4300 | Chronic | 112 | 37 ± 5.23* | 100 | 0 | 100 |
| Leon-Velarde F, et al. 2001 [54] | Cross-sectional | Peru | 4300 | Chronic | 33 | NR | 100 | 0 | 100 |
| Maignan M, et al. 2009 [48] | Cross-sectional | Peru | 4300 | Chronic | 57 | NR | 0 | 0 | 100 |
| Moraga FA, et al. 2002 [36] | Prospective cohort | Chile | 3500–4400 | Acute | | NR | - | - | - |
| Moraga FA, et al. 2008 [35] | Prospective cohort | Chile | 3500 | Acute | | 4.3 ± 1* | 43.8 | - | - |
| Peñaloza D, et al. 1963 [61] | Cross-sectional | Peru | 4540 | Chronic | 38 | 22 ± 3.8* | 0 | 100 | 0 |
| Pesce C, et al. 2005 [37] | Retrospective cohort | Argentina | 5088 | Acute | | 36.5 ± 10.1* | 15.4 | - | - |
| Quispe-Trujillo MM, et al. 2020 [50] | Cross-sectional | Peru | 5200 | Chronic | 51 | 44 ± 7* | 0 | 0 | - |
| Riaño López L, et al. 2021 [23] | Case series | Colombia | 2640 | Chronic | 6 | 11 ± 3* | 16.7 | 0 | 100 |

(*Continued*)

**Table 1.** (Continued)

| Author-Year | Study design | Country | Altitude level (m) | Type to exposure | Sample size (n) | Age (yr) | Female (%) | Native (%) | Urban (%) |
|---|---|---|---|---|---|---|---|---|---|
| Richalet JP, et al. 2002 [39] | Prospective cohort | Chile | 3800–4600 | Chronic | 29 | 25 ± 5* | 0 | - | - |
| Salazar H, et al. 2012 [25] | Cross-sectional | Peru | 3400 | Acute | | 32 (25–49) a | 55.2 | - | - |
| Seoane L, et al. 2011 [26] | Prospective cohort | Argentina | 3500–5000 | Acute | | 35 (26–44) ** | 50.4 | 87.5 | - |
| Serrano-Dueñas M. 2005 [27] | Prospective cohort | Ecuador | 4100–5800 | Acute | | 26.1 (18–41) ** | 35.7 | 0 | - |
| Siques P, et al. 2009 [28] | Prospective cohort | Chile | 3550 | Acute | | 17.9 ± 0.1* | 0 | - | - |
| Steele AR, et al. 2021 [57] | Cross-sectional | Peru | 4380 | Chronic | 24 | 40 ± 12* | 0 | 0 | 100 |
| Valencia-Flores M, et al. 2004 [51] | Prospective cohort | Mexico | 2240 | Chronic | 57 | 42.7 ± 12.1* | 59.6 | - | 100 |
| Van Roo JD, et al. 2011 [24] | Prospective cohort | Argentina | 4365–6487 | Acute | | 42.1 (39.2–45.1) ** | 9.1 | - | - |
| Vizcarra-Escobar D, et al. 2015 [56] | Cross-sectional | Peru | 4100–4300 | Chronic | 78 | NR | 0 | - | 0 |

NR: No Reported, yr (year), d (day).

* Mean age ± SD.

** Mean age (range)

a Median age (IQR)

b Age range

Native: defined as a person who has been born and raised at altitude.

Urban: defined like a place that has at least 100 houses together forming blocks or blocks and includes population centers that are district capitals.

We observed an asymmetry in the funnel plot for studies reporting the prevalence of acute and chronic mountain sickness, raising suspicions of publication bias (S4 Fig).

## Discussion

This systematic review focused on investigating the prevalence of MS in moderate to high altitude inhabitants in Latin America, which originates from a failure to adapt adequately to high altitudes and its severity is determined by hypoxemia according to the altitude and the time of exposure [40]. To our knowledge, this is the first study to report the prevalence of both acute and chronic MS along with each of their symptoms revealing significant variability in the prevalence of MS across different countries and altitudes in LATAM. There was a notable absence of data from certain countries in the region, suggesting the need for further research in those areas as also evidenced by a 40-year bibliometric analysis in which the south American countries with the highest altitude and the greatest number of publications were Peru, Bolivia and Colombia [16]. The results of this review show that both acute MS (AMS) and chronic MS (CMS) are significant issues in altitude populations in Latin America. The prevalence of AMS and CMS varied depending on altitude, country, and the presence of mild or severe symptoms. Overall, a higher prevalence of AMS was observed in Peru and Chile, while the prevalence of CMS was higher in Mexico and Colombia.

### Acute mountain sickness

Our study found that 5 out of 10 persons had AMS in LATAM countries, slightly higher occurrence compared to estimations for Europe (3 to 4 events out of 10 persons) [62]. However, our

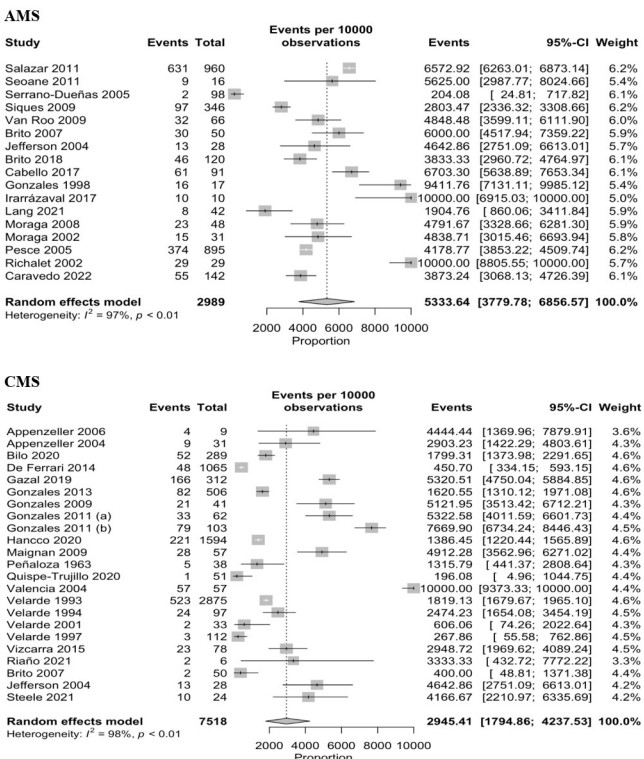

**Fig 2. Meta-analysis on the prevalence of Acute (A) and Chronic (B) Mountain Sickness.**

study involved a higher average altitude compared to the European study (3500 vs. 4,023). On the other hand, the meta-analysis included acclimatized high-altitude inhabitants of LATAM, while in Europe the majority of mountain ascents are for tourism or sport by low altitude residents [62]. AMS can manifest itself variably according to the physiological adaptation to altitude of each individual, however, a greater severity of symptoms is observed in low altitude residents who ascend to higher altitudes [63].

AMS is a complex disorder triggered by high-altitude hypoxia, initiating a pulmonary response characterized by an increase in tidal volume and respiratory rate, leading to subsequent respiratory alkalosis [63]. Simultaneously, cerebral vasodilation occurs [51], manifesting as headaches [61]. The onset of AMS typically occurs six to 12 hours after high-altitude climbing, but can manifest as quickly as within one to two hours or as late as 24 hours [64]. Given this variability, it is challenging to precisely identify the occurrences of this disease, and there is a possibility that some events may not be captured in our estimations.

The most common symptoms of AMS were headaches, while more severe complications such as pulmonary and cerebral edema, pulmonary hypertension, and polycythemia were also present, albeit to a lesser extent. These findings underscore the importance of considering a wide range of symptoms and complications when assessing the burden of MS in the region. AMS is characterized by the presence of mild symptoms [63, 65], primarily resulting from mechanisms of adaptation to hypoxemia [5, 66]. These symptoms encompass headache, fatigue, anorexia, nausea, dizziness as well as sleep disorders. Identifying the symptoms will be necessary to diagnose AMS, using the Lake Louise clinical score [64]. In our study, we identified that these symptoms were present in four of 10 people, with headaches being the most commonly reported, aligning with findings from previous reviews [67]. The occurrence of

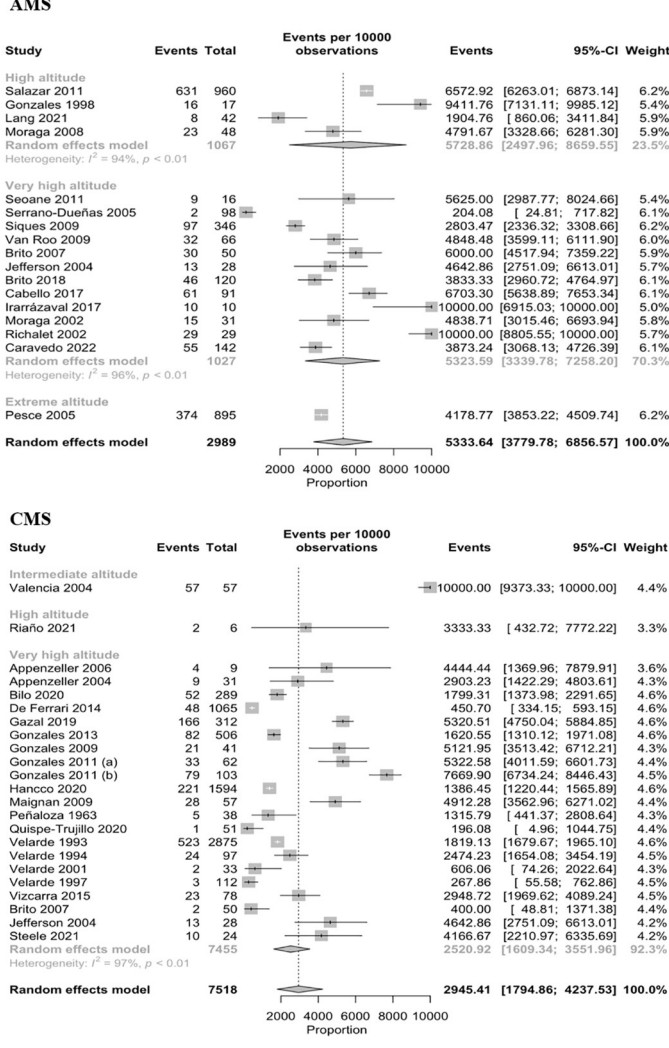

**Fig 3. Prevalence of Acute (A) and Chronic (B) Mountain Sickness by altitude.**

headaches in AMS can be explained by various mechanisms, including cerebral edema due to hypoxia, leading to increased intracranial pressure, the elevation of cerebral blood flow, difficulty in draining cerebral venous outflow and the activation of the trigeminal vascular system due to the release of nitrous oxide and vasodilation [5, 66]. Even a higher female biological susceptibility to MSA has been proposed, however, the few studies that have investigated it showed variability for [51, 64] and against it [63, 65]. Therefore, studies with larger population and methodological quality are recommended.

## Chronic mountain sickness

Lack of long-term altitude adaptation remains the cause of CMS. Our study identified that 3 out of 10 people had CMS, a significantly higher prevalence compared to the previous study conducted in Asia (1 event out of 10), which confirms that lack of adaptation is its genesis. However, despite the pathological mechanisms and the exact biological mechanism remains unknown [68], factors associated to the context such as environmental pollution, malnutrition, the lack in basic public health needs and poverty have also been described [66]. The diagnosis

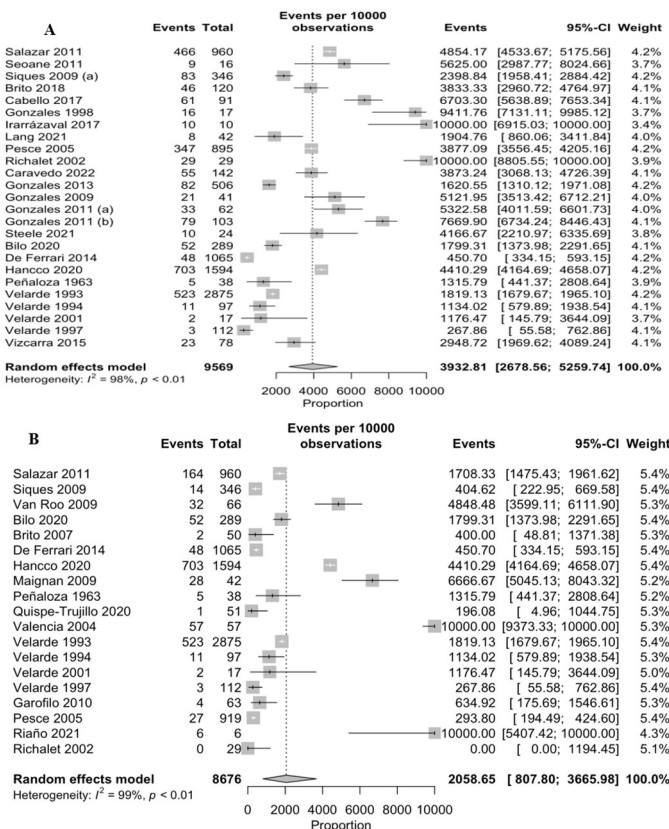

**Fig 4. Meta-analysis on the prevalence of mild (A) and severe (B) symptoms of Mountain Sickness.**

of CMS is based on the Qinghai CMS Score, which includes the evaluation of common symptoms such as shortness of breath, palpitations, sleep disturbances, heart failure, peripheral venous insufficiency, paresthesia, headache, cognitive impairment, exaggerated polycythemia and pulmonary hypertension [67].

The CMS is related with the severe symptoms, considering the mechanism involved and the chronic exposures. We identified that approximately 3 out of 10 persons has severe symptoms, higher than that reported by a study in Qinghai, China, where only 2.4% of more than 1000 residents had severe symptoms [68]. However, there are some severe symptoms such as acute pulmonary edema, that it was present in 1 event out of 10 persons, which is related to the AMS that develops approximately 4 to 12 hours after arrival [58]. The cerebral edema is also related with AMS, this has a progressive presentation and is usually associated with neurological symptoms such as decreased consciousness and/or ataxia, and can even result in coma [5, 45]. We identified lower prevalence of cerebral edema (less than 1 event out of 10 persons), but slightly lower than European estimations (1 event out of 10 persons) [46]. Other complications like exaggerated pulmonary hypertension are related to chronic exposure to altitude, being the most frequent severe symptom (2 event out of 10 persons). Our estimation differs from others reported in India where the prevalence was less than 1 event out of 10 people [33]. In relation to its origin, it is known that the chronic hypoxic stimuli of living at high altitude can cause "permanent pulmonary vascular remodeling" due to increased pulmonary vascular resistance and define a subgroup of pulmonary hypertension known as high-altitude pulmonary hypertension (HAPH) [60]. It's important to note that complicated of this can progress to

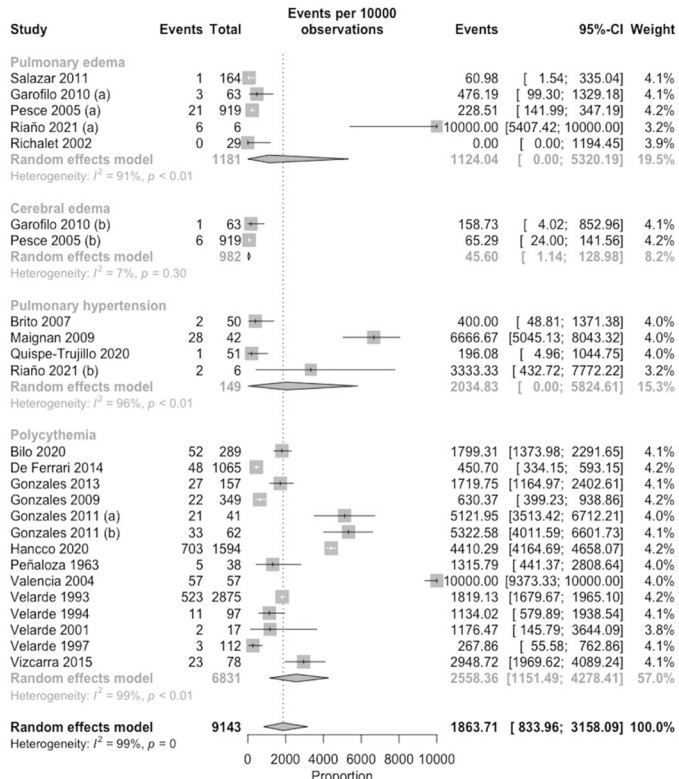

**Fig 5. Meta-analysis on the prevalence of complications of mountain sickness.**

cor pulmonale and congestive heart failure [33]. Polycythemia was presented approximately 3 events out of 10 persons. Our result was different from the study conducted in Tibetans where the prevalence was 1 event out of 10 [34]. It is possible that genetic differences influence the prevalence of polycythemia, due to the variation in the levels of soluble erythropoietin receptors (sEpoR) present in the blood of each individual, which buffers the increase in hemoglobin [52]. In addition, the existence of inherited genetic factors in susceptibility to these MSA and CMS has been suggested [69, 70], and thus there may be differences in disease prevalence between native and non-native but acclimatized residents, and it would be of interest to involve them in future research.

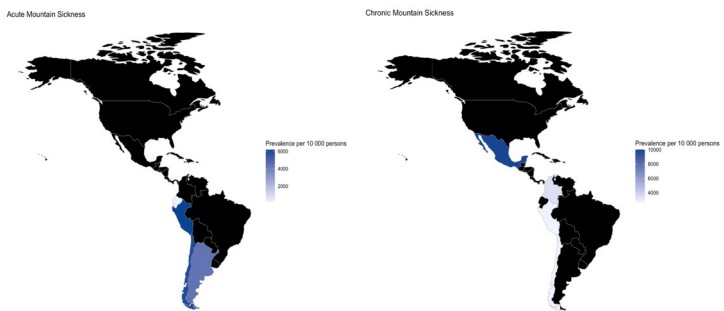

**Fig 6. Prevalence of acute and chronic mountain sickness by country, based on the number of published studies.**

## Implications of the results and recommendations for future studies

There are few systematic reviews related to altitude, in relation to MS in LATAM, no data on the prevalence of this pathology have been reported, however, the research found focuses on the general population without considering important variables such as the type of resident (immigrant or native), the generational period and the comorbidities that influence the development of the pathology, therefore it is suggested that future primary studies analyze variables such as quality of life, carry out studies in larger samples and at different altitudes. In the present review, we found a higher prevalence of AMS and CMS in LATAM high-altitude population compared to other high-altitude populations. However, it is worth recognizing that high-altitude regions present certain difficulties, including low coverage of essential primary health care services, unequal access to essential medicines, and an absence of programs for the prevention, treatment and management of MS. As a consequence, complications of the disease may present themselves earlier, in addition, an adaptive response to high-altitude with exaggerated polycythemia and pulmonary hypertension are associated with the development of arterial hypertension [53] and others such as the development of cognitive impairment. To limit the development of these, it is recommended to strengthen primary care through the use of protocols for the management of each pathology, improve the relationship between the community, health personnel and patients.

For future research, addressing these limitations by employing more rigorous study designs and including a greater number of studies with a low risk of bias is recommended. Furthermore, it would be beneficial to further explore risk factors and underlying mechanisms of MS in LATAM, as well as evaluate the effectiveness of prevention and treatment strategies. We suggest that future research considers including specific ethnic groups from the high-altitude regions in the Andes to facilitate genetic studies aimed at further understatement of their physiological responses to high altitude. Finally, the results presented are intended to generate adequate epidemiological information.

## Limitations and strengths of the review

Limitations in this review should be noted. First, the prevalence estimates were not standardized due to the different sample sizes of each study and the statistical methods used. Additionally, there was great heterogeneity between the studies reviewed, thus limiting their evidence certainty, and some studies did not have the necessary information to be include in the meta-analysis or to conduct subgroup analyses. Most of the studies had small sample size, which generated imprecision about individual studies that generated a low evidence certainty and a high risk of bias identified in most of the included studies, which could affect the validity and generalizability of the results. Similarly, a potential bias is the poorly defined target population in some studies, with a bias towards AMS due to variability regarding the inclusion of climbing tourists with AMS or the exclusion of long-term high-altitude residents. Finally, evidence of publication bias was observed, suggesting that studies with negative results may not have been published.

The study has some strengths as the first systematic review and meta-analysis on the prevalence of MS in LATAM that has the Andes as an altitudinal group, therefore the data presented can serve as background for future studies and the systematized information can be useful for researchers, public health professionals working in high altitude regions and the development of national strategies for the prevention and treatment of AMS and CMS in mid and high-altitude regions. Also, we address a clear gap in the literature when considering high-altitude residents in LATAM for presenting unique characteristics such as shorter residence time at altitude.

## Conclusion

This systematic review provides an overview of the prevalence of AMS in altitude inhabitants in LATAM which occurs in 5 out of 10 people at high altitude, while CMS occurs in 3 out of 10. The most frequent symptom type was polycythemia and the least frequent was cerebral edema. The results underscore the importance of addressing this public health issue and highlight the need for further research including more participants and data from the different Andean countries to obtain more representative results and a better understanding of its epidemiology and address its clinical and public health implications in the region.

## Supporting information

**S1 Checklist. PRISMA checklist.**
(DOCX)

**S1 Table. Excluded studies.**
(DOCX)

**S2 Table. Search strategy.**
(DOCX)

**S1 Fig. Risk of bias of included studies.**
(DOCX)

**S2 Fig. Sensitivity analysis on prevalence of Acute (A) and Chronic (B) Mountain Sickness between risk of bias.**
(DOCX)

**S3 Fig. Sensitivity analysis on prevalence of Acute (A) and Chronic (B) Mountain Sickness excluding outliers.**
(DOCX)

**S4 Fig. Publication bias on the prevalence of acute (A) and chronic (B) mountain sickness.**
(DOCX)

## Author Contributions

**Conceptualization:** J. Pierre Zila-Velasque, Pamela Grados-Espinoza.

**Data curation:** J. Pierre Zila-Velasque, Pamela Grados-Espinoza, P. Alejandra Goicochea-Romero, Gustavo Tapia-Sequeiros, J. Enrique Pascual-Aguilar, Arturo J. Ruiz-Yaringaño, Shamir Barros-Sevillano, Jhon Ayca-Mendoza.

**Formal analysis:** Wendy Nieto-Gutierrez.

**Funding acquisition:** Wendy Nieto-Gutierrez.

**Investigation:** J. Pierre Zila-Velasque, Pamela Grados-Espinoza, P. Alejandra Goicochea-Romero, Gustavo Tapia-Sequeiros, J. Enrique Pascual-Aguilar, Arturo J. Ruiz-Yaringaño, Shamir Barros-Sevillano, Jhon Ayca-Mendoza, Wendy Nieto-Gutierrez.

**Methodology:** J. Pierre Zila-Velasque, Pamela Grados-Espinoza, P. Alejandra Goicochea-Romero, Gustavo Tapia-Sequeiros, J. Enrique Pascual-Aguilar, Arturo J. Ruiz-Yaringaño, Shamir Barros-Sevillano, Jhon Ayca-Mendoza.

**Project administration:** J. Pierre Zila-Velasque, Pamela Grados-Espinoza, Wendy Nieto-Gutierrez.

**Resources:** J. Pierre Zila-Velasque, P. Alejandra Goicochea-Romero, Wendy Nieto-Gutierrez.

**Software:** Wendy Nieto-Gutierrez.

**Supervision:** J. Pierre Zila-Velasque, P. Alejandra Goicochea-Romero, J. Enrique Pascual-Aguilar, Shamir Barros-Sevillano, Wendy Nieto-Gutierrez.

**Validation:** J. Pierre Zila-Velasque, P. Alejandra Goicochea-Romero, Gustavo Tapia-Sequeiros, J. Enrique Pascual-Aguilar, Shamir Barros-Sevillano, Jhon Ayca-Mendoza.

**Visualization:** J. Pierre Zila-Velasque, Pamela Grados-Espinoza, P. Alejandra Goicochea-Romero, Gustavo Tapia-Sequeiros, J. Enrique Pascual-Aguilar, Arturo J. Ruiz-Yaringaño, Shamir Barros-Sevillano, Jhon Ayca-Mendoza, Wendy Nieto-Gutierrez.

**Writing – original draft:** J. Pierre Zila-Velasque, Pamela Grados-Espinoza, P. Alejandra Goicochea-Romero, Gustavo Tapia-Sequeiros, J. Enrique Pascual-Aguilar, Arturo J. Ruiz-Yaringaño, Shamir Barros-Sevillano, Jhon Ayca-Mendoza.

**Writing – review & editing:** J. Pierre Zila-Velasque, Pamela Grados-Espinoza, P. Alejandra Goicochea-Romero, Gustavo Tapia-Sequeiros, J. Enrique Pascual-Aguilar, Arturo J. Ruiz-Yaringaño, Shamir Barros-Sevillano, Jhon Ayca-Mendoza, Wendy Nieto-Gutierrez.

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
