## [Decision Letter · Decision Letter 0]

10 Mar 2024

PONE-D-23-42448Mountain sickness in altitude inhabitants of Latin America: a systematic review and meta_analysisPLOS ONE

Dear Dr. Nieto-Gutierrez,

Thank you for submitting your manuscript to PLOS ONE. After careful consideration, we feel that it has merit but does not fully meet PLOS ONE’s publication criteria as it currently stands. Therefore, we invite you to submit a revised version of the manuscript that addresses the points raised during the review process.

We look forward to receiving your revised manuscript.

Kind regards,

Esteban Ortiz-Prado

Academic Editor

PLOS ONE

Journal Requirements:

**Additional Editor Comments:**

Dear Authors,

After a thorough evaluation by peer reviewers, the consensus is that your manuscript could make a valuable contribution to the field. However, there are several areas that require revision to meet the journal's standards for publication. The reviewers have provided detailed feedback, highlighting the need for clarification in certain sections, consistency in terminology, and corrections for typographical errors. Furthermore, enhancements in the discussion and conclusion sections are recommended to strengthen the impact of your findings.

Please find attached the reviewers' comments for your consideration. We invite you to submit a revised version of your manuscript that addresses these points. Alongside the revised manuscript, please include a response letter detailing how you have addressed each comment provided by the reviewers.

Your revised manuscript will undergo further review to ensure that all concerns have been adequately addressed. We believe that your study holds significant potential, and we look forward to your revised submission.

Should you have any questions or require clarification on the feedback provided, please do not hesitate to contact us.

Thank you for considering [Journal Name] for your work. We look forward to your response.

Sincerely,

Esteban Ortiz

Reviewers' comments:

Reviewer's Responses to Questions

**Comments to the Author**

1. Is the manuscript technically sound, and do the data support the conclusions?

Reviewer #1: Partly

Reviewer #2: Partly

2. Has the statistical analysis been performed appropriately and rigorously? 

Reviewer #1: Yes

Reviewer #2: Yes

3. Have the authors made all data underlying the findings in their manuscript fully available?

Reviewer #1: Yes

Reviewer #2: Yes

4. Is the manuscript presented in an intelligible fashion and written in standard English?

Reviewer #1: Yes

Reviewer #2: Yes

5. Review Comments to the Author

Reviewer #1: GENERAL COMMENTS

I consider the review/meta-analysis study to be adequate and provides an interesting overview to date on the prevalence of mountain sickness (acute and chronic types) in Latin American countries. However, the study suffers, in my opinion, from the analysis of some aspects that I consider interesting, and some sections of the manuscript should be made more solid. There are also some typos that should be corrected and minor changes that I propose that should be taken into account. I specify all of this, point by point, in the following sections:

SPECIFIC COMMENTS

ABSTRACT:

It states that the information on acute and chronic disease only comes from the Himalayas, but this is not true, as there are numerous studies done in the Andes, as shown in this meta-analysis. I suggest the following “...but the information mostly comes from the eastern continent (Himalayas)…”.

It is noted that polyglobulia is a “pathology” and below it appears as a “symptom”, and cerebral edema as a “symptom”. These are an error that should be corrected, as polyglobulia per se is a physiological response to hypoxia, and can only be considered pathological if it is excessive for a given altitude. Likewise, cerebral edema is a physiological alteration and a clinical picture, not a symptom.

I suggest deleting the last sentence: “Therefore, we recommend…”

INTRODUCTION:

In my opinion this section is somewhat poor in illustrating the subject. It should be slightly enriched.

Line 92: in this paragraph it should not be said that there are no studies that do not include information from the Andes, but rather that the information is comparatively scarce with other mountainous places throughout the world.

Line 94: “…Nepal Himalya and Ethiopian”. Put it this way: “…Nepal Himalaya and Ethiopian highlands” (I suggest adding also the reference: Xing et al. Adaptation and mal-adaptation to ambient hypoxia; Andean, Ethiopian and Himalayan patterns. PLoS One, 2008).

Line 98: “pulmonary hypertension” is not a complication but a physiological response of the organism to the hypoxia of altitude; the complication would be an “exaggerated pulmonary hypertension”...

The bibliography in this Introduction section should be enriched (I suggest referencing here some articles, such as Monge 1942, Hancco et al. 2020 [already in the reference list], Champigneulle et al. 2023, Waterlow and Bunjé 1966, Sánchez et al. 2022, etc., as well as some others that appear in the Supplementary material-3). At least this would further illustrate the Introduction section, even if there are articles that are then excluded in the meta-analysis.

METHODS:

Line 113: define the acronym PECO

Lines 116-117: use the acronyms AMS and CMS.

Line 123: perhaps mention should be made here to “mountain sickness” and make reference to the Supplementary material-2. The keyword “Andes” would also have been useful in the search.

Line 185: “pulmonary hypertension, and polycythemia” should only be considered pathological responses if they are excessive for a given altitude (I think this should be stated in this paragraph).

Lines 195-200: total sample population included (10,549) does not match the sum of AMS (3,087) and CMS (7,448). Is it a mistake, or what is the reason? If it is a misinterpretation on my part, it is possible that other readers may have the same misinterpretation as me.

Lines 199-200: “Regarding the studies that evaluated AMS, most of the studies were from Chile (n = 8 studies) and Peru/Argentina (n = 4 studies)…”. Do these refer to studies that included larger samples?

Lines 223-225: Data from AMS (5,334/10,000) and CMS (2,945/10,000) are not consistent with those shown in lines 197-198: AMS (3,087/10,549) and CMS (7,448/10,549). The authors should clarify this apparent discrepancy, so that there are no errors in the interpretation of the data shown in two different places in this Results section (these discrepant figures may lead to significant confusion in the interpretation of the data by readers).

DISCUSSION:

I suggest that the authors rework this section. I propose that you make a more in-depth analysis and focus on the context of the results obtained through the extensive review/meta-analysis performed, and make less continuous allusion to the pathophysiological aspects of MS and its different forms of presentation (AMS, CMS) and derived clinical complications (cerebral edema, pulmonary edema, etc.). I also suggest that the definitions of AMS and CMS should appear succinctly in the Introduction section and less so in the Discussion. In my opinion, the entire Discussion section lacks sufficient narrative strength and makes for uninspiring reading, but I think that especially the first part of this section (lines 286-341) is somewhat convoluted in terms of the concepts presented. Therefore, I encourage you to make an effort to rewrite this section to improve it, and thus achieve better quality of the manuscript and not detract from the value and effort that has gone into the research of the study, as reflected in the Material and Results sections.

I also add the following specifications:

Lines 287-290: Given that the meta-analysis includes acclimatized high-altitude inhabitants of LATAM, this high percentage of AMS is striking compared to the prevalence in Europe (where the majority of mountain ascents are for tourism or sport by low altitude residents). The authors could make a small allusion to this detail, despite the fact that the average altitude reached in the studies analyzed is comparatively only 500 meters higher. Perhaps, in order to give more solidity to the comparative data with Europe, a reference to some other studies on the subject carried out in European mountains could be added here.

Line 290: the physiological explanations here are very succinct “Physiologically,…”. I suggest to the authors a sentence like this: “AMS is a complex disorder triggered by high-altitude hypoxia and can present with diverse clinical manifestations”.

Line 301: “dizziness” is missing, as well as it would be appropriate to name (and include a reference to) the Lake Louise clinical score, which is the most widely used for the diagnosis of AMS.

Line 305: also add “difficulty in draining cerebral venous outflow”.

Line 308: “Lack of adequate adaptation to altitude remains the cause of CMS”. To avoid getting into interpretative differences between ‘acclimatization’ and ‘adaptation’, I suggest that for the majority of readers it should be specified as such: “long-term adaptation”.

Line 311: I suggest including here the most typical manifestations of CMS (exaggerated polycythemia and pulmonary hypertension, heart failure, fatigue, peripheral venous insufficiency, headache and cognitive impairment, etc.) as well as naming the Qinghai Score (include reference) used for the diagnosis and clinical evaluation of CMS.

Line 316: The study by Richalet et al. (ref. 72) is not on CMS, and is compared here with this present study!

Line 329-330: the “pulmonary hypertension” perhaps it should be expressed here as an “exaggerated pulmonary hypertension”.

Line 332: “personas” (correct to English).

Line 335: define HAPH acronym (high-altitude pulmonary hypertension).

Line 344: also use here the acronym MS (“…mountain sickness…”).

Line 344: In my opinion, it lacks some further analysis of AMS and CMS in native and non-native highlanders (*).

Line 350: use acronyms AMS and CMS.

Line 354: use acronym MS.

Line 356: “poliglobulia and pulmonary hypertension are considered as pathologies associated…”. I suggest writing it in the following way: “an adaptive response to high-altitude with exaggerated polyglobulia and pulmonary hypertension are associated…”

Line 366: use acronyms “…MS in LATAM…”

(*) a) Regarding of AMS, one aspect that I think it would be very interesting to analyze is why high-altitude residents (non-native but have been acclimatized to hypoxia for at least 12 months) develop this type of mountain sickness. Is it because they ascend or move to areas of even higher altitude than their usual altitude residence? And even, are there differences in the prevalence of AMS between Andean natives and non-natives?

(*) b) Regarding the CMS, are there differences in the prevalence of CMS between Andean natives and non-natives residents who have been living at high altitude for years? Table 1 shows the percentages of the samples with exclusively Andean natives (8 studies), exclusively non-natives residents (13 studies), and both types (3 studies). Likewise, it would be interesting to analyze the probable percentage difference in CMS between men and women (Table 1 shows 8 studies that include samples of both sexes in chronic exposure).

I also believe it would be very interesting to analyze all these aspects (a and b) and you could even add those statistical data in the RESULTS section. In the event that it is impossible to extract or deduce the data I propose (on the published studies included in the review/meta-analysis), however, the authors should also refer to these aspects in the Discussion and this section would be more enriched.

CONCLUSSION:

It is correct, but in my opinion it is very brief.

TABLE 1: in the first column “et al.” should be specified in studies with multiple authors. Likewise, at the end of each author’s name I suggest including the corresponding reference number supraindexed (that would facilitate the search in the bibliographic list).

Typographical errors:

Garofilo (incorrect), Garófoli (correct); Valencia (incorrect), Valencia-Flores (correct); Penaloza (incorrect), Peñaloza (correct); Riano (incorrect), Riaño (correct); Salazar 2011 (incorrect), Salazar 2012 (correct).

FIGURES 2 and 3: In these 2 figures I would recommend specifying the acronyms AMS and CMS, instead of “A” and “B”, as in my opinion it would be more illustrative. Also correct it when the respective Figures are mentioned in the text.

FIGURES 2, 3, 4 and 5: I suggest that “et al.” be included in those studies carried out by more than 1 author.

FIGURE 6: in the wording “Prevalence of Acute and Chronic Mountain Sickness by country”, it should be specified that it is the “reported prevalence based on published studies”. As the legend goes, it seems that in Mexico (in its population residing at high altitude) there is more susceptibility to CMS than in Peru, for example, which is hard to believe as true.

REFERENCES LIST:

There are references that are repeated in the list: e.g. ref. 5 and 8 (note this when referenced in the text).

Some journal names are not abbreviated correctly.

Some of the articles’ statements appear with initial capital letters and others do not (to unify the criteria according to the journal’s standards).

Ref. 24: the name of the first author is “van Roo” (match with Table 1).

Ref. 27: the journal’s name is spelled incorrectly. I think it is “Cephalalgia”.

Ref. 31: the journal’s name is spelled incorrectly. I think it is “Medicina (B Aires)”.

Ref. 52: the journal’s name is spelled incorrectly. I think it is “Int J Obes Relat Metab Disord”.

SUPPLEMENTARY MATERIAL. FIGURES 5 and 6: I suggest including “et al.” in those studies performed by more than one author. Likewise, in these 2 figures I would recommend specifying the acronyms AMS and CMS, instead of “A” and “B”, as in my opinion it would be more illustrative.

Reviewer #2: This paper is an attempt to resolve the global discrepancies in study and publication regarding a condition which mainly occurs in the lower income regions of the worlds and hence a good and deserving effort.

Manuscript needs to be read by native English speaker.

Abstract: better define dwellers at high altitude in the abstract. It only requires a few words and is done in Methods, but raises some confusion as to who is your target population: long term residents as well as people in transit such as tourists?

Also for limitations sector it is wise to stress your awareness of the fact that inclusion or exclusion of tourist climbers with AMS potentially distorts your findings towards AMS.

Line 125 l197 says 'y' instead of and; Dominican Republic?

Polyglobulia is used initially, then polycythemia vera and eventually in line 356 it is again polyglobulia. Advice is to stick to polycythemia and define both better in case you intend to use it distinctly.

From line 79 perhaps choose to omit a.s.l. as this is obvious from that point onwards.

L 184-186 outcome measures pulmonary oedema cerebral oedema polycythemia and pulmonary hypertension need to be defined according to international standards (references have been given).

l 114-116 and results section: define the altitude ranges and classes in Methods.

l 281-282 A potential bias is the ill defined target population as stated above with a bias towards AMS in populations which exlude the long term dwellers on altitude. This may have to be stressed a bit more firmly here.

L 287-289, 315-316, 328-329 perhaps good to describe your denominator better than ** out of ** people: what is the target population you base the statistics on? (If you define it once, you do not have to do it again)

Finally as a recommendation based on your study you could indicate that the inclusion of certain ethnic groups would call for genetic studies of such populations in order to further investigate their physiology on high altitude. After all the Andean high altitude population is rather unique in their habitat!

6. PLOS authors have the option to publish the peer review history of their article (what does this mean?). If published, this will include your full peer review and any attached files.

Reviewer #1: No

Reviewer #2: **Yes: **MCJ Dekker

---

## [Author Response · Author response to Decision Letter 0]

29 May 2024

Dear Editor and reviewers

Thanks for reviewing our manuscript and giving us an opportunity to improve it. Please find enclosed a rebuttal letter containing the responses point by point to each of the recommendations/observations, as well as the corrected and revised manuscript. 

Response to recommendations: 

Reviewer #1: 

Comment 1: It states that the information on acute and chronic disease only comes from the Himalayas, but this is not true, as there are numerous studies done in the Andes, as shown in this meta-analysis. I suggest the following “...but the information mostly comes from the eastern continent (Himalayas)…”.It is noted that polyglobulia is a “pathology” and below it appears as a “symptom”, and cerebral edema as a “symptom”. These are an error that should be corrected, as polyglobulia per se is a physiological response to hypoxia, and can only be considered pathological if it is excessive for a given altitude. Likewise, cerebral edema is a physiological alteration and a clinical picture, not a symptom.

I suggest deleting the last sentence: “Therefore, we recommend…”.

Response 1: We agree with the reviewer’s observation; the wording has been corrected in the manuscript as shown below: 

“…but most of the available information comes from the eastern continent (Himalayas), …”

“… The most common physiological alteration was polycythemia (2,558 events), while cerebral edema was the less common (46 events). Clinical conditions were more prevalent at high altitudes for both types of MS.”

Deleted the last sentence: “Therefore, we recommend…”

Comment 2: In my opinion this section is somewhat poor in illustrating the subject. It should be slightly enriched. Line 92: in this paragraph it should not be said that there are no studies that do not include information from the Andes, but rather that the information is comparatively scarce with other mountainous places throughout the world.

Response 2: We agree with the reviewer’s observation; this section has been corrected according to the reviewer's suggestions as shown below. 

“…. However, these reports are mainly from eastern areas (such as Nepal Himalaya and Ethiopian highlands) with comparatively scarce information with other mountainous places in the world, such as the Andes 11.” 

Comment 3: Line 94: “…Nepal Himalya and Ethiopian”. Put it this way: “…Nepal Himalaya and Ethiopian highlands” (I suggest adding also the reference: Xing et al. Adaptation and mal-adaptation to ambient hypoxia; Andean, Ethiopian and Himalayan patterns. PLoS One, 2008).

Response 3: We agree with the reviewer’s observation; the reference has been added in the manuscript as shown below. 

“… such as Nepal Himalaya and Ethiopian highlands)”

References:

11. Xing G, Qualls C, Huicho L, et al. Adaptation and Mal-Adaptation to Ambient Hypoxia; Andean, Ethiopian and Himalayan Patterns. PLOS ONE. 2008;3(6):e2342. doi:10.1371/journal.pone.0002342

Comment 4: Line 98: “pulmonary hypertension” is not a complication but a physiological response of the organism to the hypoxia of altitude; the complication would be an “exaggerated pulmonary hypertension”...

Response 4: We agree with the reviewer’s observation; the wording has been corrected in the manuscript as shown below. 



“……, and complications (exaggerated pulmonary hypertension, pulmonary and cerebral edema).

Comment 5: The bibliography in this Introduction section should be enriched (I suggest referencing here some articles, such as Monge 1942, Hancco et al. 2020 [already in the reference list], Champigneulle et al. 2023, Waterlow and Bunjé 1966, Sánchez et al. 2022, etc., as well as some others that appear in the Supplementary material-3). At least this would further illustrate the Introduction section, even if there are articles that are then excluded in the meta-analysis.

Response 5: We agree with the reviewer’s suggestion; the references has been added in the manuscript as shown below. 

“Previous studies have reported the frequency of MS in different regions of the world 8,9,10. However, these reports are mainly from eastern areas (such as Nepal Himalaya and Ethiopian highlands) with comparatively scarce information with other mountainous places in the world, such as the Andes 11. In fact, a previous systematic review 12 determined a global prevalence of MS at approximately 12%; however, they omitted to include studies from LATAM 13–15.”

References: 

13. Hancco I, Bailly S, Baillieul S, et al. Excessive Erythrocytosis and Chronic Mountain Sickness in Dwellers of the Highest City in the World. Front Physiol. 2020;11:773. doi:10.3389/fphys.2020.00773

14. Champigneulle B, Brugniaux JV, Stauffer E, et al. Expedition 5300: limits of human adaptations in the highest city in the world. J Physiol. 2023;n/a(n/a). doi:10.1113/JP284550

15. Monge C. Life in the Andes and Chronic Mountain Sickness. Science. 1942;95(2456):79-84. doi:10.1126/science.95.2456.79

Comment 6: Line 113: define the acronym PECO.

Response 6: We agree with the reviewer’s observation and suggestion; we have added the components of the PECO acronym as shown below. 

“….. the following PECO (where the means of P: population E: exposure, O: outcome) structure: 1) population: altitude inhabitants (defined as a person who live in a place more than 12 months); 2) exposure: altitude ( following the ranges of intermediate altitude 1500-2500 m, from where physiological changes are detectable), high altitude (2500-3500 m), very high altitude (3500-5800 m), extreme altitude (>5800 m.), and “death zone” (>8000 m.a.s.l.) 20); and 3) outcome: prevalence of AMS and CMS, according to each disease and study”

Comment 7: Lines 116-117: use the acronyms AMS and CMS.

Response 7: We agree with the reviewer’s observation and suggestion; the wording has been corrected in the manuscript as shown below. 

“…..outcome: prevalence of AMS and CMS, according to each disease and study”

Comment 8: Line 123: perhaps mention should be made here to “mountain sickness” and make reference to the Supplementary material-2. The keyword “Andes” would also have been useful in the search.

Response 8: We agree with the reviewer’s observation and suggestion; the wording has been corrected in the manuscript as shown below. 

“…. using keywords related to “mountain sickness” and “Latin America”. Also, we include terms for each Latin America country that report a high altitude (Peru, Chile, Argentina, Bolivia, Venezuela, Ecuador and Colombia), taking into account all the cities located at high altitudes. The search strategy of each database is available in Supplementary 2”

On the other hand, although the term “Andes” might have been useful in the search, we have added reference terms for each Latin American country that reports high altitudes, so it is unlikely that the term “Andes” would have affected the search results.

Comment 9: Line 185: “pulmonary hypertension, and polycythemia” should only be considered pathological responses if they are excessive for a given altitude (I think this should be stated in this paragraph).

Response 9: We agree with the reviewer’s observation and suggestion; the wording has been corrected in the manuscript as shown below. 

“……, and complications (pulmonary edema, cerebral edema, exaggerated pulmonary hypertension, and exaggerated polycythemia).”

Comment 10: Lines 195-200: total sample population included (10,549) does not match the sum of AMS (3,087) and CMS (7,448). Is it a mistake, or what is the reason? If it is a misinterpretation on my part, it is possible that other readers may have the same misinterpretation as me.

Response 10: We agree with the reviewer’s observation. There was an error in the wording, the population of AMS was 2,945. CMS was 7,448 and two studies with both populations of 156, which now add up correctly. The wording has been corrected in the manuscript as shown below. 

“…. Of the 39 studies, acute mountain sickness (AMS) was evaluated in 15 studies 24–38 (n=2 945), chronic mountain sickness (CMS) in 22 13,23,39–58 (n=7 448), and both in 2 59,60 (n=156).”

Comment 11: Lines 199-200: “Regarding the studies that evaluated AMS, most of the studies were from Chile (n = 8 studies) and Peru/Argentina (n = 4 studies)…”. Do these refer to studies that included larger samples?

Response 11: Thank you for your comment. It actually refers to the countries that contributed the most number of the studies. 

“….. Regarding the studies that evaluated AMS, most of the studies were from Chile (n = 8 studies) and Peru/Argentina (n = 4 studies), which are the countries that contributed the most research on the subject.”

Comment 12: Lines 223-225: Data from AMS (5,334/10,000) and CMS (2,945/10,000) are not consistent with those shown in lines 197-198: AMS (3,087/10,549) and CMS (7,448/10,549). The authors should clarify this apparent discrepancy, so that there are no errors in the interpretation of the data shown in two different places in this Results section (these discrepant figures may lead to significant confusion in the interpretation of the data by readers).

Response 12: The discrepancy in the sum of the previously included sample has been clarified and corrected. The data from AMS (5,334/10,000) and CMS (2,945/10,000) mentioned by the reviewers refers to the prevalence (not to the sample) calculated per 10,000 people. 

Comment 13: I suggest that the authors rework this section. I propose that you make a more in-depth analysis and focus on the context of the results obtained through the extensive review/meta-analysis performed, and make less continuous allusion to the pathophysiological aspects of MS and its different forms of presentation (AMS, CMS) and derived clinical complications (cerebral edema, pulmonary edema, etc.). I also suggest that the definitions of AMS and CMS should appear succinctly in the Introduction section and less so in the Discussion. In my opinion, the entire Discussion section lacks sufficient narrative strength and makes for uninspiring reading, but I think that especially the first part of this section (lines 286-341) is somewhat convoluted in terms of the concepts presented. Therefore, I encourage you to make an effort to rewrite this section to improve it, and thus achieve better quality of the manuscript and not detract from the value and effort that has gone into the research of the study, as reflected in the Material and Results sections.

Response 13: We agree with the reviewer’s observation and suggestion. We have decided to rewrite all the discussion. You will be able to see the changes in the manuscript.

Comment 14: Lines 287-290: Given that the meta-analysis includes acclimatized high-altitude inhabitants of LATAM, this high percentage of AMS is striking compared to the prevalence in Europe (where the majority of mountain ascents are for tourism or sport by low altitude residents). The authors could make a small allusion to this detail, despite the fact that the average altitude reached in the studies analyzed is comparatively only 500 meters higher. Perhaps, in order to give more solidity to the comparative data with Europe, a reference to some other studies on the subject carried out in European mountains could be added here.

Response 14: We agree with the reviewer’s observation; we have added information on the topic in the manuscript as shown below.

“….. the meta-analysis included acclimatized high-altitude inhabitants of LATAM, while in Europe the majority of mountain ascents are for tourism or sport by low altitude residents 99. AMS can manifest itself variably according to the physiological adaptation to altitude of each individual, however, a greater severity of symptoms is observed in low altitude residents who ascend to higher altitudes 100.

Comment 15: Line 290: the physiological explanations here are very succinct “Physiologically,…”. I suggest to the authors a sentence like this: “AMS is a complex disorder triggered by high-altitude hypoxia and can present with diverse clinical manifestations”.

Response 15: We agree with the reviewer’s observation; the wording has been corrected in the manuscript as shown below.

“…. AMS is a complex disorder triggered by high-altitude hypoxia, initiating a pulmonary response characterized by an increase in tidal volume and respiratory rate, leading to subsequent respiratory alkalosis 64”

Comment 16: Line 301: “dizziness” is missing, as well as it would be appropriate to name (and include a reference to) the Lake Louise clinical score, which is the most widely used for the diagnosis of AMS.

Response 16: We agree with the reviewer’s observation; the wording has been corrected and we added information in the manuscript as shown below. 

“…..These symptoms encompass headache, fatigue, anorexia, nausea, dizziness as well as sleep disorders. Identifying the symptoms will be necessary to diagnose AMS, using the Lake Louise clinical score.”

Comment 17: Line 305: also add “difficulty in draining cerebral venous outflow”.

Response 17: We agree with the reviewer’s observation; the wording has been corrected in the manuscript as shown below. 

“……leading to increased intracranial pressure, the elevation of cerebral blood flow, difficulty in draining cerebral venous outflow and the activation of the trigeminal vascular system due to the release of nitrous oxide and vasodilation 5,69.”

Comment 18: Line 308: “Lack of adequate adaptation to altitude remains the cause of CMS”. To avoid getting into interpretative differences between ‘acclimatization’ and ‘adaptation’, I suggest that for the majority of readers it should be specified as such: “long-term adaptation”.

Response 18: We agree with the reviewer’s observation; the wording has been corrected in the manuscript as shown below. 

“Lack of long-term altitude adaptation remains the cause of CMS…”

Comment 19: Line 311: I suggest including here the most typical manifestations of CMS (exaggerated polycythemia and pulmonary hypertension, heart failure, fatigue, peripheral venous insufficiency, headache and cognitive impairment, etc.) as well as naming the Qinghai Score (include reference) used for the diagnosis and clinical evaluation of CMS.

Response: We agree with the reviewer’s observation; we added information in the manuscript as shown below. 

“The diagnosis of CMS is based on the Qinghai CMS Score, which includes the evaluation of common symptoms such as shortness of breath, palpitations, sleep disturbances, heart failure, peripheral venous insufficiency, paresthesia, headache, cognitive impairment, exaggerated polycythemia and pulmonary hypertension 105.”

Comment 20: Line 316: The study by Richalet et al. (ref. 72) is not on CMS, and is compared here with this present study!

Response 20: We agree with the reviewer’s observation; we have added new information and replaced the previously mentioned reference as shown below.

“… We identified that approximately 3 out of 10 persons has severe symptoms, higher than that reported by a study in Qinghai, China, where only 2.4% of more than 1000 residents had severe symptoms 106.”

Comment 21: Line 329-330: the “pulmonary hypertension” perhaps it should be expressed here as an “exaggerated pulmonary hypertension”.

Response 21: We agree with the reviewer’s observation; the wording has been corrected in the manuscript as shown below.

“Other complications like exaggerated pulmonary hypertension are related to chronic exposure …..”

Comment 22: Line 332: “personas” (correct to English).

Response 22: We agree with the reviewer’s observation; the wording has been corrected in the manuscript as shown below.

“Our estimation differs from others reported in India where the prevalence was less than 1 event out of 10 people 76.”

Comment 23: Line 335: define HAPH acronym (high-altitude pulmonary hypertension).

Response 23: We agree with the reviewer’s observation; the wording has been corrected in the manuscript as shown below. 

“……. vascular resistance and define a subgroup of pulmonary hypertension known as high-altitude pulmonary hypertension (HAPH) 77.”

Comment 24: Line 344: also use here the acronym MS (“…mountain sickness…”).

Response 24: We agree with the reviewer’s observation; the wording has been corrected

---

## [Editor Report · Decision Letter 1]

4 Jun 2024

Mountain sickness in altitude inhabitants of Latin America: a systematic review and meta_analysis

PONE-D-23-42448R1

Dear Dr. Nieto-Gutierrez,

We’re pleased to inform you that your manuscript has been judged scientifically suitable for publication and will be formally accepted for publication once it meets all outstanding technical requirements.

Kind regards,

Esteban Ortiz-Prado

Academic Editor

PLOS ONE
---

## [Editor Report · Acceptance letter]

20 Jul 2024

PONE-D-23-42448R1 

PLOS ONE

Dear Dr. Nieto-Gutierrez, 

I'm pleased to inform you that your manuscript has been deemed suitable for publication in PLOS ONE. Congratulations! Your manuscript is now being handed over to our production team.

Kind regards, 

on behalf of

Dr. Esteban Ortiz-Prado 

Academic Editor

PLOS ONE